# Age-Friendly Transportation Policies and Practices in the U.S. and China: A Comparative Study

**Xueming Chen** [1,*] and **Suwei Feng** [2]

1  L. Douglas Wilder School of Government and Public Affairs, Virginia Commonwealth University, Richmond, VA 23284, USA
2  School of Public Economics and Administration, Shanghai University of Finance and Economics, Shanghai 200433, China; fsuwei@mail.shufe.edu.cn
*  Correspondence: xchen2@vcu.edu

**Abstract:** Despite significant political, cultural, and socioeconomic differences, the U.S. and China share some common development issues, including, but not limited to, an aging population, deficient age-friendly transportation systems, and insufficient government funding support to address the issues faced by their transportation-disadvantaged populations (TDP). Through an extensive literature review, this paper first evaluates and compares the major TDP-related laws, regulations, transportation service delivery procedures, and existing age-friendly transportation improvements made in the U.S. and China. Next, it highlights the differences between the U.S. and China in their socioeconomic characteristics and the implications these have on transportation modal shares and age-friendly transportation planning priorities. In the concluding section, it summarizes the research findings and makes preliminary recommendations.

**Keywords:** aging; age-friendly transportation; public transit; paratransit; the U.S. and China

## 1. Introduction

It is well known that the world's population is aging. Virtually every country in the world is experiencing growth in the number and proportion of older adults in their population due to longer life expectancy and declining fertility rates. In 2019, the global population aged 65 and over was 703 million (9% of the total), a number which is projected to reach 1.5 billion (16% of the total) by 2050 [1,2]. According to the official UN definition, any society with more than 7% of the population aged 65 and over or 10% of the population aged 60 and over is called an aging society [3]. Based on this definition, the entire world has already entered into an aging society.

Consistent with this worldwide trend, the United States (U.S.) and China have undergone the following population aging trajectories:

- According to decennial population census data, the U.S. became an aging society as early as the 1940s, with the population aged 65 and over accounting for 8.2% of the total population by 1950 [4]. This proportion reached 16.8% in 2020 [5]. Older adults represent the fastest-growing segment of the U.S. population. According to 2010 U.S. census data, the population aged 65 and older grew 15% from 2000 to 2010, compared with growth of about 10% in the overall population during the same period.

- Around the year 2000, China also became an aging society. According to the 5th National Population Census, China's population aged 65 and above reached 88.11 million (approximately 7% of the total) in 2000 [6]. This proportion quickly increased to 14.2% in 2021 [7]. Therefore, there is a time lag of approximately 50 years between the U.S. and China in terms of the time of entering into an aging society. Overall, the existing Chinese age structure is still younger than its American counterpart.

While bringing new opportunities, population aging will also create tremendous challenges in such areas as labor markets, social welfare, pension systems, built environment/community redesign, housing, transportation, and many others.

Using the U.S. and China as empirical cases, this paper primarily focuses on age-friendly transportation, which is intertwined with transportation for individuals with disabilities and low-income groups due to their data overlapping.

The reasons why the U.S. and China are picked for comparison in this paper include, but are not limited to, the following:

First, so far, no study has been conducted to comprehensively compare and evaluate the transportation policies and practices for the transportation-disadvantaged populations (TDP) between the U.S. and China. This paper intends to fill this research gap, with an emphasis on age-friendly transportation policies and practices. The U.S. is the largest developed, private automobile-dominant country and China is the largest developing, slow, or public transportation-dominant country in the world. They may well represent the two ends of the transportation modal spectrum, between which many other countries may lie. Due to this reason, other countries may take lessons from the age-friendly transportation planning experience from the U.S. and Chinese cases in combination.

Second, on the issue of aging, the U.S. and China have comparable backgrounds and face similar issues despite the 50-year time lag. Both countries paid attention to the transportation equity issue, which requires the equal treatment of older adults regarding their mobility.

Third, since the U.S. has entered the aging society and passed aging-, disability-, and other related laws much earlier than China, the U.S. presumably has much more established and mature policies for transportation-disadvantaged populations. In the meantime, China's aging process seems to be occurring at a faster rate than those of other countries and has distinguishing characteristics in terms of its older population size and modal share.

Fourth, data availability is another major reason for picking the U.S. and Chinese cases due to the authors' research bases in the U.S. and China, respectively.

It should be pointed out that, in the U.S., older adults belong to one subcategory of the broadly defined TDP category, along with two other subcategories including individuals with disabilities and low-income persons. The U.S. TDP-related transportation policies and practices introduced in this paper thus pertain to all three TDP subcategories and are not limited to the older adults subcategory. The U.S. Government Accountability Office (GAO) has published a series of reports on TDP topics, especially on transportation program coordination issues and strategies [8–17].

The three central research questions explored in this paper are: What have been the U.S. and Chinese TDP-related transportation policies and practices? What are their similarities and differences? In what aspects can the U.S. and China learn from each other in building and improving their age-friendly transportation systems?

In terms of the research methodology, this comparative study is based on an extensive literature review and qualitative data analysis. It primarily follows the logical process of transportation project development from policies to programs, and finally to projects.

Following this introduction, this paper consists of five interrelated sections: Section 2 reviews and compares the TDP-related Laws and Regulations in the U.S. and China; Section 3 describes the TDP-related transportation service delivery procedures in the U.S. and China; Section 4 gives a succinct introduction into the local age-friendly transportation improvements in the U.S. and China; Section 5 discusses the differences between the U.S. and China in terms of their socioeconomic characteristics and assesses the implications of these differences on their modal shares and age-friendly transportation policies and practices; Section 6 summarizes the research findings and makes preliminary recommendations.

## 2. TDP-Related Laws and Regulations in the U.S. and China

The U.S. and China have very different political and legal systems. The U.S. is a federal representative democracy. Under this check-and-balance structure, the legislative branch (the U.S. Congress at the federal level) makes laws, the executive branch (different federal departments) enforces laws through issuing administrative regulations, and the judicial branch (the U.S. Supreme Court) interprets the laws to ensure their constitutionality. The boundaries between the three branches are clear and distinct.

China, on the other hand, is a unitary state under the leadership of the Communist Party of China (CPC). While the National People's Congress passes laws, the instructions, suggestions, or opinions from the Central Committee of the CPC (sometimes jointly issued with the State Council) often have quasi-legal effects as well. So, the distinction between laws and regulations in China is somewhat blurred.

Irrespective of their differences, both countries have enacted TDP-related laws and promulgated administrative regulations under different contexts. Transportation policies, programs, and funding authorizations all initially start from national transportation laws and regulations, and then follow the top-down paths for local implementation.

### 2.1. TDP-Related Laws in the U.S. and China

### 2.1.1. U.S. TDP-Related Laws

Our literature review suggests that the following federal laws passed by the U.S. Congress since the 1960s are directly or indirectly related to TDP-related transportation.

In our opinion, the first law was the *Civil Rights Act of 1964*. Driven by the motive of promoting racial equity, the U.S. Congress enacted the landmark *Civil Rights Act of 1964*. Title VI of this law prohibits any discrimination based on race, color, and national origin in programs and activities receiving federal financial assistance. Due to the protection of minority groups by Title VI, the federal government has been subsidizing the operation of public transportation, on which minority groups heavily rely in their travels. Due to this reason, the racial-equity driven *Civil Rights Act of 1964* indirectly benefits the entire transportation-disadvantaged population.

The second law was the *Older Americans Act of 1965 (OAA)*, which was directly related to TDP transportation. OAA was enacted to provide services to older adults to help them remain in their homes and communities for as long as possible. Examples of services are transportation for older adults with and without mobility impairments. OAA established authority for grants to states for community planning and social services, research and development projects, and personnel training in the field of aging. At the federal level, OAA established the Administration on Aging (AoA) to administer the newly created grant programs and to serve as the focal point on matters concerning older adults. Collectively, the current national aging services network comprises 56 State Agencies on Aging, 622 Area Agencies on Aging, and more than 260 Title VI Native American aging programs. Further, the network is supported by tens of thousands of service providers and volunteers [18].

As the grantees of OAA's Title III B funds, local area agencies on aging (which are typically non-profit organizations) provide different types of supportive services to local older adults and caregivers. Take Senior Connections (The Capital Area Agency on Aging) in Richmond, Virginia, for example. Senior Connections provides different types of support for older adults to stay home: Home Delivered Meals Program, Friendship Cafés, Telebridges Telephone Reassurance Program, Personal Care, Money Management, and Ride Connection. Its Ride Connection service helps older adults and individuals with disabilities with transportation needs (e.g., Friendship Café visits and medical appointments) [19].

Directly related to TDP transportation, the third law was the *Rehabilitation Act of 1973*, which covers federal contractors and programs receiving federal funds. As a result of Section 504 of this law, paratransit began to be provided by non-profit human service agencies and public transit agencies. The federal law prohibited the exclusion of the disabled from "any program or activity receiving federal financial assistance". The Federal

Transit Administration defined requirements for making buses accessible or providing complementary paratransit services within public transit service areas.

Also directly related to TDP transportation, the fourth law was the *Americans with Disabilities Act (ADA) of 1990*. Without directly providing funding support, the *Americans with Disabilities Act (ADA) of 1990* required public transit agencies that provide a fixed-route service to provide a "complementary paratransit" service to people with disabilities who otherwise cannot use the fixed-route bus or rail service because of a disability. The ADA-related federal regulations specifically identify a population of disabled customers who are entitled to this service as a civil right.

Concerning the minimum service characteristics that must be met, an ADA complementary paratransit service must be provided within the 3/4 mile of a fixed bus route or rail station, during the same hours and days, with a fare no more than twice the regular fixed route fare. While the transit agency is required to provide paratransit for trips with origins and destinations within the 3/4 mile of a route/station, paratransit-eligible customers who are outside the service area could still use the service if they can get themselves into the service area.

After 1990, there were other TDP transportation-related laws passed by the U.S. Congress. For example, the *1991 Intermodal Surface Transportation Efficiency Act (ISTEA)* and *1998 Transportation Equity Act for the 21st Century (TEA-21)* appropriated sizeable amounts of funds to provide driver training, development of intelligent transportation systems (ITS), and other public transit facilities. The *Older Americans Act Amendments of 1992 and 2006* further reaffirmed the critical importance transportation plays in elderly health and also demanded that each state set aside enough funds for elderly persons' transportation projects.

With the above brief review, it appears that there is no U.S. law that comprehensively and specifically addresses all TDP transportation issues at once.

### 2.1.2. Chinese TDP-Related Laws

In China, under the leadership of the Central Committee of the Communist Party of China, the National People's Congress is responsible for making laws, some of which are related to undertakings for the aged. The law-making process for older adults accelerated since the 1990s.

The first Chinese TDP-related law was the *Law of the People's Republic of China on the Protection of Disabled Persons: 1990*. Article 33 of this law stipulates to provide disabled persons with preferential treatment and assistance.

The second Chinese TDP-related law was the *Law of the People's Republic of China on Protection of the Rights and Interests of the Elderly: 1996*. This landmark law played an important role in safeguarding elderly rights and interests, fostering the undertakings of the elderly, and promoting harmonious social development. The elderly referred to in this law are Chinese citizens at or above the age of 60. This important law was amended multiple times, in 2009, 2012, 2015, and 2018. It now has nine chapters and 85 articles.

Article 58 of the latest law amendment, effective from 1 July 2013, stipulates that urban public transportation, highway, water transportation, and aviation should provide preferential treatment to older adults. Article 64 requires the construction and provision of accessible transportation facilities.

Unfortunately, the amended version of this law did not have many detailed provisions on transportation services for TDP.

As summarized in Table 1, both the U.S. and China do not have laws that comprehensively address all TDP transportation issues. Racial equity has a lower priority in China because even though China has 56 nationalities, Han nationality alone accounts for about 92% of the Chinese population. Due to this reason, China does not have many Civil Rights-related laws passed.

**Table 1.** Summary of Selected U.S. and Chinese Laws.

| Laws | The U.S. | China | Comments |
|---|---|---|---|
| Civil Rights Act of 1964 | Anti-racial discrimination law | | Indirectly related to TDP, no funding support. |
| Older Americans Act of 1965 (OAA) | Providing comprehensive services for older adults | | Title III B funds provide funding support to local agencies on aging. |
| Rehabilitation Act of 1973 | Anti-disability discrimination law | | As a result of Section 504 of this law, paratransit began to be provided by non-profit human service agencies and public transit agencies. |
| Americans with Disabilities Act (ADA) of 1990 | Anti-disability discrimination law | | All federally subsidized public transit operators are required to provide paratransit services. |
| People's Republic of China on the Protection of Disabled Persons: 1990 | | Anti-Disability Law | Not related to transportation. |
| Law of the People's Republic of China on Protection of the Rights and Interests of the Elderly: 1996 | | Similar to OAA in the U.S. | Not related to transportation. |

*2.2. TDP-Related Regulations in the U.S. and China*

2.2.1. U.S. Regulations

At the federal level, U.S. regulations primarily take the form of Presidential Executive Orders (EO) and departmental rules.

There were two TDP-related EOs issued by former President George W. Bush:

- The first one was *EO 13217 (Community-Based Alternatives for Individuals with Disabilities),* issued on 18 June 2001. This EO directed all federal agencies to support new public transportation services and public transportation alternatives for individuals with disabilities to foster their independence and participation in the community.
- The second one was *EO 13330 (Human Service Transportation Coordination),* issued on 24 February 2004. *EO 13330* was created to enhance access to transportation to improve mobility, employment opportunities, and access to community services for persons who are transportation-disadvantaged. Part of this new coordination initiative created the Interagency Transportation Coordinating Council on Access and Mobility (CCAM) consisting of multiple federal agencies, such as the Department of Transportation (DOT), Department of Health and Human Services (HHS), and Department of Labor (DOL), etc.

On public transportation, the departmental rules were primarily issued by the Federal Transit Administration (FTA) under the Department of Transportation (DOT). Tapping into the private sector, the FTA 5310 program (49 U.S.C. 5310) provides formula funding to states to assist private non-profit groups in meeting the transportation needs of elderly adults and people with disabilities when the transportation service provided is unavailable, insufficient, or inappropriate to meet these needs. In other words, it provided capital investment and "nontraditional" investment beyond the complementary paratransit services provided by the *Americans with Disabilities Act (ADA).*

FTA 5317 (New Freedom program) provides new public transportation services to overcome existing barriers facing Americans with disabilities seeking integration into the workforce and full participation in society, while expanding the transportation mobility

options available to persons with disabilities beyond the requirements of the *Americans with Disabilities Act of 1990* (42 U.S.C. 12101, et seq.).

The FTA Circular 9045.1, effective from May 1, 2007, incorporates provisions of the *Safe, Accountable, Flexible, and Efficient Transportation Equity Act: A Legacy for Users (SAFETEA-LU)*. This requires projects selected for funding to be derived from a locally developed, coordinated public transit-human services transportation plan (Coordinated Plan) and that the plan be developed through a process that includes representatives of a public, private, and non-profit transportation and human service providers and participation by members of the public.

### 2.2.2. Chinese Regulations

To implement the Chinese laws, the Chinese State Council and its ministries put forward a series of aging plans and regulations, as shown in Table 2.

**Table 2.** Selected Chinese Aging Plans and Regulations.

| Issuing Date | Policy Name | Issuing Agency | Main Contents |
|---|---|---|---|
| 27 February 2006 | Opinions on strengthening the grassroots-level aging undertakings | National Committee on Aging | Increase welfare benefits for the aged |
| 17 September 2011 | 12th five-year plan of the Chinese elderly undertaking development | State Council | Emphasize home-based care services and establish and refine county, town, and community service networks |
| 27 October 2016 | 13th five-year plan for accessible environment construction | China Disabled Persons' Federation, etc. | Set targets for accessible environment construction |
| 22 January 2018 | Implementation opinion of strengthening and improving transportation-disadvantaged populations' travel services | Ministry of Transport and other related entities | Build an accessible system by the year 2020 and improve the accessible travel service level |
| 21 November 2019 | Medium and long-range plans facing population aging | Central Committee of CPC and State Council | Guide Chinese aging undertakings for 2035 and 2050 |
| 24 November 2020 | Notice to solve the elderly difficulty in using smart technologies | Office of the State Council | Help seniors use smart technologies |
| 9 December 2020 | Notice on constructing pilot projects for age-friendly communities | National Health Commission and National Aging Commission | Upgrade community capacity and level to better serve old adults |
| 20 October 2021 | Notice on naming the age-friendly communities | National Health Commission | Announce the naming of 992 communities as 2021 age-friendly communities |
| 18 November 2021 | Opinions on strengthening the aging undertakings in the new era | Central Committee of CPC and State Council | Guide on establishing and improving several aspects of the aging undertakings |
| 1 September 2023 | Law of the People's Republic of China on Constructing the Accessible Environment | Standing Committee of the National People's Congress | Strengthen the construction of accessible environments to better serve the aged |

As a result of the above efforts in China, the functions of aging care through family and employers have been weakened. In the meantime, the functions played by the government and society have been strengthened.

In a bid to improve age-friendly transportation development, China's Ministry of Transport (MOT), Ministry of Housing and Urban-Rural Development, State Railway Administration, China Civil Aviation Bureau, State Post Office Bureau, China Disabled Persons'

Federation, and Office of the National Committee on Aging jointly published the *Implementation Opinion of Strengthening and Improving Transportation-Disadvantaged Populations' Travel Services* in 2018.

This *Implementation Opinion* was issued with the understanding that China at that time had 230 million elderly persons aged 60 and above, and 85 million disabled persons, making the issue of providing accessible travel services for transportation-disadvantaged populations a top priority. The initial goal was to build an accessible system by the year 2020 and improve the accessible travel service level.

More specific targets of this very important *Implementation Opinion* are to:

(1) Provide accessible facilities in all newly built or renovated railway stations, highway service plazas, second class and above bus stations, urban ferries, international passenger water ports and airports, and urban subway stations; have the Post Office deliver all mail, printed matters, and remittance notices according to their mailing addresses; encourage those cities with suitable conditions to purchase low-floored buses; suggest that all cities with a population of more than 5 million people should operate 100% low-floored buses;

(2) Establish a fully covered, seamlessly integrated, safe, and comfortable accessible travel service system by the year 2035. This system should constantly be improved to adequately satisfy the travel needs of transportation-disadvantaged populations;

(3) Incorporate all accessible transportation infrastructure plans into local development plans. The construction must be undertaken in an orderly way; and

(4) Upgrade travel service quality through adopting innovative service models, establishing a travel information service system, improving service levels, and ensuring travel safety.

### *2.3. Summary of U.S. and Chinese Laws and Regulations*

The above literature review reveals the following facts:

First, both the U.S. and China do not have comprehensive laws that directly address all TDP-related transportation issues at once.

Second, only a few regulations directly deal with TDP-related transportation issues. In the U.S., the leading agency is the Federal Transit Administration (FTA) for public transportation. In China, they are the Ministry of Transport and other related central government agencies.

Third, in the U.S., many human and social service agencies provide incidental transportation services for their clients. In China, aging undertakings are primarily home-based, supplemented by social and other institutional agencies, which focus on elderly care services and accessible environment construction, with little or no transportation services provided.

### 3. TDP-Related Transportation Service Delivery Procedures in the U.S. and China

Closely related to the preceding section, this section gives an introduction to the transportation service delivery procedures in the U.S. and China, highlighting their respective top-down paths of transportation funds and local project implementation.

### *3.1. Service Delivery Procedures in the U.S.*

In the U.S., the federal and state governments are responsible for setting policies and allocating funds. As grantees, the local government agencies or non-profit organizations are responsible for implementing planning measures to serve the transportation-disadvantaged populations. Local grantees may deliver transportation services either directly or indirectly using contracting with private transit providers or providing transit passes, taxi vouchers, mileage reimbursement to program participants, or some combination of these methods. Some programs may use federal funds to purchase and operate their vehicles.

According to Rosenbloom (1982), the U.S. policy for the elderly and handicapped has been made in two largely distinct program arenas. The first involves the policies and

directives of the U.S. Department of Transportation (DOT). The second is the human and social service network. The human and social service program expenditures [administered by the Department of Health and Human Services (HHS)] far outstrip DOT's direct expenditures on programs [administered by the Federal Transit Administration (FTA)] for these citizens [20]. The HHS and FTA procedures are described at length below:

First, the *Department of Health and Human Services (HHS) dictated procedures*: The legal foundation of the HHS procedures is Title III Part B of the *Older Americans Act of 1965 (OAA)*. The grants for supportive services are allocated by the Administration on Aging (AoA) within the HHS to the state units on aging, which further allocate the funds to local Area Agencies on Aging (AAAs) that either directly provide transportation services or contract with local service providers.

In addition, HHS also has other programs directly or indirectly related to TDPs, such as the Community Services Block Grant Programs, Medicaid, Rural Health Care Services Outreach Program, etc.

Second, the *Federal Transit Administration (FTA) dictated procedures*: Of about 80 TDP-related federal programs, the FTA administers 7 programs that support TDP transportation [14]. For example, FTA's Enhanced Mobility of Seniors and Individuals with Disabilities program (Enhanced Mobility program, or 5310 Program) provides formula funding to states to serve the special needs of transit-dependent populations beyond traditional public transportation services. The *Moving Ahead for Progress in the 21st Century Act (MAP-21) of 2012* expanded the eligibility of the Enhanced Mobility program to include activities previously eligible under the New Freedom program (5317 Program) and to public transportation projects that improve access to fixed-route services and decrease reliance by individuals with disabilities on complementary paratransit. The Enhanced Mobility program requires grantees to develop coordinated public transit-human services plans. In addition, the FTA also has other TDP-related programs, such as the Capital and Training Assistance Program for Over-the-Road Bus Accessibility, Capital Investment Grants (Section 5309), Job Access and Reverse Commute, Urbanized Area Formula Program (Section 5307), etc. Many programs use existing public or private transportation services through such methods as contracting for services with private transportation providers, or through providing bus tokens, transit passes, taxi vouchers, or mileage reimbursement to volunteers or program participants [14].

Third, the *Interagency Transportation Coordinating Council on Access and Mobility (CCAM) dictated procedures*: As a hybrid option, the CCAM procedures created the United We Ride (UWR) initiative to facilitate coordination between transportation and human services programs. Developed by DOT, HHS, DOL, and the Department of Education (DOE), "United We Ride" is a five-part transportation coordination initiative officially launched in December 2003. These five components are the framework for action, state leadership awards, the national leadership forum on human services transportation coordination, state coordination grants, and help along the way [10].

As a result of the UWR, state and local officials made tremendous efforts in coordinating councils, regional and local planning, one-call centers, mobility managers, and vehicle sharing [10].

### 3.2. Service Delivery Procedures in China

According to the China Ministry of Civil Affairs, by the first quarter of 2022, China had 360,000 aging agencies and facilities, which provided 8.126 million beds for the aged. Between 2012 and 2021, the Chinese government invested 35.9 billion Chinese RMB in elderly undertakings and facility constructions [21]. Like the U.S., some Chinese aging agencies may occasionally provide incidental transportation services (like pick-up and drop-off) to their aging clients but most do not.

**4. Age-Friendly Transport Improvements in the U.S. and China**

The U.S. and Chinese service delivery procedures introduced in Section 3 primarily touch on the public transportation services provided by transit operators and the incidental transportation services provided by human and social service agencies. This federally-focused picture is certainly incomplete.

Both the U.S. and China have made significant age-friendly transportation improvements in other areas, especially at local levels.

*4.1. Age-Friendly Transportation Improvements in the U.S.*

In the U.S., many age-friendly transportation improvements have been proposed or implemented by local and state transportation agencies.

On the highway side, here are some examples:

- Improving roadway and traffic sign designs to attract elderly drivers' attention. Traffic warning signs are placed on crosswalks to alert motorists and pedestrians to pay attention to transportation safety, and special audible or countdown signals at intersections are installed so older adults can hear and see. Pedestrian refuge islands are constructed in the middle of wide streets to improve elderly safety.
- Implementing volunteer driver programs to give rides to their neighboring older adults. Social and human service agencies reimburse the auto mileage provided by the volunteers.
- Adjusting intersection traffic signal timing to match elderly walking speed.
- Tightening the driver's license renewal rules for older drivers by the state Departments of Motor Vehicles (DMVs). In Virginia, drivers aged 75 and older must renew their driver's licenses every five years in person, along with a vision test. In California, due to its heavier traffic and higher driving hazards, the age limit is further lowered to 70 years old. The driver's license renewal must be done in person, along with both knowledge and vision tests.

On the public transit side, some examples of improvements are:

- Deploying wheelchair-accessible ramps for buses. Alternatively, buses have installed lift equipment, which makes it easier for elderly riders to board.
- Transit companies provide simple and convenient timetables and route maps at every transit stop so that elderly and disabled passengers can read them. Bus stops need to improve their lighting conditions and provide benches and all-weather facilities.
- Ensuring that bus drivers receive adequate training so they know the demands of elderly and disabled passengers, speak politely, and allow elderly and disabled passengers to use accessible facilities.

On the paratransit side, following the ADA mandates, all public transit operators receiving federal subsidies must provide complementary door-to-door or curb-to-curb paratransit services (vans or small buses) to eligible riders within ¾ mile of fixed bus routes. In the Richmond, Virginia (RVA) region, the Greater Richmond Transit Company (GRTC) provides paratransit services to eligible riders who are either disabled or 80 years and older. Paratransit also has other beneficial effects. Blumenberg and Manville (2004) also recognize that demand-responsive transit, though expensive per ride, shows early promise in poverty reduction because it can provide a door-to-door service rivaling that of the automobile, requires less capital outlay than bus or rail, and has operational flexibility [22].

*4.2. Age-Friendly Transport Improvements in China*

On the highway side, the Chinese government began to allow older adults aged 70 or older to apply for their driver's licenses starting on 22 October 2020. The elderly driver's license application process still involves tests of memory, cognition, and reaction time, as well as a physical examination [23]. In addition, some streets in China also have underpasses or audible traffic signals for the convenience of elderly street-crossing.

On the public transit side, most Chinese cities provide discounted or free bus passes to older adults. Starting in 2016, Shanghai provided an age-based cash allowance to eligible older adults so they have their discretion in making their bus-boarding decisions. In addition, some bus operators also provide accessible wheelchair ramps for the convenience of older adults to board buses.

On the walking side, street furniture for walking is still lacking in most Chinese cities. Even though most Chinese city streets have sidewalks, crosswalks, overpasses, and underpasses, there are very few benches, water fountains, restrooms, and trash receptacles provided along the streets. Some streets are very wide and their traffic signals are not timed for the walking pace of older adults to cross streets.

### 4.3. Remaining Issues of Age-Friendly Transportation in the U.S. and China
#### 4.3.1. The U.S.

Through efforts made during the past decades, U.S. transportation policies for transportation-disadvantaged populations have achieved a certain degree of success. For example, almost all cities' public entities, private non-profit organizations, hospitals, faith-based organizations, and other civil organizations have provided various types of incidental senior transportation facilities. A few ethnically complex cities also provide multilingual services. For example, the Los Angeles-based paratransit company *Access* provides English/Spanish bilingual services. Many cities' buses were also equipped with accessible facilities, which made it more convenient for elderly and disabled people to board and alight transit vehicles.

Despite the above achievements, U.S. transportation policies for transportation-disadvantaged populations still have many deficiency issues yet to be resolved:

First, the federal government has not provided enough funds for aging and disabled undertakings, which has made it difficult for many transit companies to provide and expand their expensive ADA paratransit services. The paratransit services should also lower the disability requirements to make more elderly riders eligible.

Second, the coordination between public transit and human service transportation remains insufficient, which creates fragmentation, waste, and inefficiency.

Third, the federal monitoring of public transit companies largely remains a formality, lacking a rigorous quality checking of the transit operating data submitted by public transit companies, which has made the National Transit Database somewhat inaccurate.

#### 4.3.2. China

In China, except for a few cities such as Beijing (which is the first Chinese city to provide good paratransit services via transit agencies), most Chinese cities do not have the widespread application of paratransit services for the elderly and disabled. Unlike the ADA in the U.S., Chinese laws have not legally mandated the provision of complementary paratransit services for the elderly and disabled yet. Chinese older adults are accustomed to using the old way of hailing taxi cabs by hand on the streets, rather than using smartphones due to their unfamiliarity with new internet technology.

Several Chinese cities have tried to use taxi cabs to provide door-to-door transport services to the elderly and disabled, similar to the *Access* paratransit services in Los Angeles. However, the results were not satisfactory due to low utilization rates, parking difficulties, and a lack of incentives for drivers to pick up older adults or individuals with disabilities. The co-author of this paper previously used Weixin v7.0.10 software to interview several transportation officials on the taxi services in their cities in 2019. The interview results are summarized below:

- *Shenzhen*: The city had 96,880 dial-a-ride taxi cabs, of which only 100 taxi cabs were specifically designed for disabled persons. Therefore, most taxi cabs were not equipped with wheelchair-accessible facilities.
- *Xi'an*: The pilot project had 50 wheelchair-accessible vehicles. However, these specially designed vehicles were underutilized, thus yielding poor economic benefits. Recently,

these vehicles seemed to be converted back to regular taxi cabs, which were not wheelchair-accessible vehicles. This case suggests that, unless the government can provide financial subsidies, taxi drivers are not incentivized to provide services to the elderly and disabled because it is difficult and time-consuming to pick up and drop off the elderly and disabled. If the taxi fare for the elderly and disabled remains the same as that of normal passengers, taxi drivers would have lower incentives to do so. A government subsidy to incentivize taxi cab drivers is key.

- *Beijing*: The Beijing Disabled Persons Federation subsidizes and provides certain incentives to taxi cabs serving disabled persons. The fares of the Beijing accessible taxi cabs are determined by the market and posted on the platform. For those online hailing taxi cabs serving first and second-class disabled passengers, the Beijing Disabled Persons Federation will apply for additional funds to provide subsidies.
- *Shanghai*: Shanghai also tried to provide taxi services to disabled persons. However, the results are less desirable due to low utilization rates.

On the pedestrian walking side, most Chinese cities generally lack good street furniture, such as benches, trash receptacles, water fountains, etc. Even bus stops do not have good amenities.

## 5. Discussion on U.S.-China Differences and Their Implications: A Recap

### 5.1. U.S. and China Differences

Aside from their huge differences in political and legal systems, the U.S. and China also have fundamental differences in socioeconomic characteristics, vehicle ownership, transportation modal shares, etc.

Some facts are highlighted below:

In 2023, the population density of China was 148 persons/km$^2$, but the population density in the U.S. was only 37 persons/km$^2$. The former is four times as high as the latter [24].

In 2022, China had a per capita Gross Domestic Product (GDP) of $12,720.2 (current US$), whereas the U.S. had $76,329.6 per capita GDP. The former is only one-sixth of the latter [25].

In 2021, 1000 American people had over 800 vehicles, but 1000 Chinese people had fewer than 200 vehicles [26].

These huge differences have significant implications.

### 5.2. Implications on Their Elderly Modal Shares

Due to the above huge differences, the U.S. and China have different modal shares for elderly persons.

In the U.S., the private vehicle continues to be the primary transportation mode for older adults after their retirement. Public transit has a negligible modal share for American elderly persons. Even non-drivers over age 65 took only 8 percent of all trips via public transit but took 66 percent of all their trips in a private car [27].

In contrast, Chinese older adults primarily rely on walking or public transit as their travel modes. According to Mao and Ren (2005), many surveyed Chinese cities' elderly walking modal shares were well above 40%, even close to 80%. Their public transit modal shares were typically between 10% and 30% [28]. Cheng (2020) conducted a similar survey in Beijing. She found that among the elders of the 55–65-year-old group, walking (73%) was the most popular travel mode [29].

### 5.3. Implications on Their Age-Friendly Transportation Planning

In the U.S., continued automobility is apparent for elderly Americans in the future. Therefore, most efforts are expected to be made to improve elderly driving safety and comfort. Typically, planning measures are roadway design improvement, street furniture, traffic signs, driver training, elderly driver license renewal, etc.

In addition, due to the legal mandates, public transit and paratransit services in the U.S. will continue to be provided and improved despite their lower modal shares. Accessibility, convenience, affordability, and safety are key considerations of public transit and human service agencies. The successes of these programs are ultimately contingent upon sustained government funding support and commitment.

In China, accessible public transportation and a pedestrian-friendly walking environment are top priorities for age-friendly transportation planning.

Mao and Huang (2006) made four suggestions to improve China's age-friendly transportation systems: (1) strengthen the construction of public transportation facilities to make sure they are elderly-accessible and user-friendly; (2) pay attention to the accessible design of transportation systems; (3) improve the pedestrian-friendly walking environment; (4) strengthen elderly driver safety training and tighten the elderly driver license renewal process [30].

From the perspective of Chen (2018), a direct cash allowance provided to elderly bus riders is most efficient and may also reduce unnecessary bus travel [31].

The Chinese Democratic League (2023) made several suggestions for improving China's Age-Friendly Transportation Systems: (1) improve the national standards and codes regarding age-friendly and accessible transportation systems; (2) construct slow transportation and pedestrian-friendly walking environments for the aged; (3) upgrade elderly trip service quality, reduce the digital divide, and develop demand-responsive public transportation for the aged; (4) build age-friendly urban transportation information service platforms to guide elderly travel [32].

### 5.4. In What Aspects can the U.S. and China Learn from Each Other?

China can learn from the U.S. in enacting more detailed and enforceable laws and regulations, providing paratransit services and volunteer driver programs, etc.

The U.S. can also learn from China in higher implementation efficiency and stronger intergovernmental coordination.

### 6. Summary of Findings and Conclusions

The entire world is aging. Both the number and proportion of older adults are increasing in both developed and developing nations. While healthcare and the fiscal challenges of the growing aging population have been well recognized and thus have attracted the attention of the public, the transportation implications of aging societies are also far-reaching and have recently become a topic of growing interest in planning and research [33–35].

Consistent with the worldwide trend, both the U.S. and China have entered aging societies. Each country needs to adopt and implement suitable age-friendly transportation policies to fit their circumstances.

In the U.S., private vehicles will continue to be the dominant travel mode for older adults after their retirement. To accommodate their travel demand, government and transportation agencies have to improve the safety of the roadway system so it will be more user-friendly for senior American citizens. In the meantime, due to various federal mandates, public transit will continue to be provided for both captive and choice riders irrespective of their income, ethnicity, and other socioeconomic status. For those persons with disabilities, transit operators will provide complementary paratransit services. The continued provision and improvement of incidental transportation, public transit, and paratransit services for American senior citizens requires more governmental funding support, intergovernmental coordination, and public–private partnerships.

In China, accessible public transportation, paratransit, pedestrian-friendly walking environments, and taxi cab services should be the top priorities in age-friendly transportation planning. Chinese elders used to rely on their children to provide transportation, but the government needs to be responsible for providing more paratransit (especially taxi cab services) to them and incentivize taxi cab drivers to do so.

Despite their priority differences, one commonality exists between the U.S. and China. In both countries, government funding support is critical to the success of building age-friendly transportation systems. Whether it be improving accessible public transit, expanding ADA paratransit services, retrofitting pedestrian facilities in the U.S., or providing discounted or free bus passes and incentivizing taxi drivers to serve more elderly and disabled in China, all interventions require substantial government funding support and subsidies. All these funding commitments should be codified in laws and regulations.

There are two major limitations of this paper that we would like to touch on. First, it does not have detailed data for a quantitative analysis. Second, it lacks first-hand survey data. Therefore, this literature review-based paper primarily provides a qualitative analysis. Nevertheless, we hope that this comparative paper on the elderly transportation planning experience from the U.S. and China has takeaway points that will benefit other countries in the future.

**Funding:** The first and second authors would like to thank the National Natural Science Foundation of China for financial support, Grant No. 71774133 and Grant No. 71871131, respectively.

**Institutional Review Board Statement:** Not applicable.

**Informed Consent Statement:** Not applicable.

**Data Availability Statement:** Not applicable.

**Conflicts of Interest:** The authors declare no potential conflicts of interest concerning the research, authorship, and/or publication of this article.

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
