# Peer review of "Age-Friendly Transportation Policies and Practices in the U.S. and China: A Comparative Study"

_sustainability, doi:10.3390/su16020921_

Round 1

Reviewer 1 Report

Comments and Suggestions for Authors

The introduction provides sufficient background and includes all relevant references, however, it is important to review the U.S. and Chinese laws, regulations, and policies related to transportation- 85 disadvantaged populations

-All cited references are relevant to the research, it need to be improved (citation forms, include access date).

-The research design is adequate

The methods are adequately described, and only the rules and regulations for the case studies are discussed.

-The results are scarcely presented in a "clear" way, the relevance of analyzing and comparing the regulations is not indicated, but there is a proposal or redesign to propose guidelines or some more substantial design proposal.

-The conclusions need to be backed up by the results.

In the proposed approach of the case study, unclear methods are used, it seems to be only analytical as a tool for sustainability assessment. However, the study requires a major revision in the revision of the indicators and substantial improvements.

The study requires correction. There are some errors in the text.

Recommended articles:

1.     Mengzhu Zhang, Pengjun Zhao, Literature review on urban transport equity in transitional China: From empirical studies to universal knowledge, Journal of Transport Geography, Volume 96, 2021, 103177, ISSN 0966-6923, https://doi.org/10.1016/j.jtrangeo.2021.103177. (https://www.sciencedirect.com/science/article/pii/S0966692321002301)

2.     Lee, J., Vojnovic, I., & Grady, S. C. (2018). Los "desfavorecidos por el transporte": forma urbana, género y viajes en automóvil frente a viajes no automovilísticos en la región de Detroit. Estudios Urbanos55(11), 2470-2498. https://doi.org/10.1177/0042098017730521

The reviewer proposed checking the sentence between 382 and 415 line numbers.

It is suggested to delve deeper into the 569 implementation parameters

The critical review of the literature and the need for the study must be carefully explained, the application of case studies is mentioned, and the following articles are suggested that will contribute to a better case study.

-It lacks sufficient background applied in the field and not only literature review as bibliographic references, but it also mentions applied antecedents, however, the type of practices as a resource is not detailed.

- The narrative should be more cohesive, and the overall ideas should be better integrated. Background to similar examples should be shown. In addition, the problem and objectives of the research should be clarified in this section. The context section should be in the results or materials and methods section if this is the case in a scientific article.

-The figure captions must be completed, based on the data available to you (Source: author, date).

Comments on the Quality of English Language

Extensive editing of English language required

Author Response

Dear Reviewer: Thank you so much for your excellent comments.

First, we have reviewed the key U.S. and Chinese laws, regulations, and policies directly relevant to Transportation Disadvantaged Populations.

Second, the citation forms have been improved, including authors, publication dates, website addresses, and access dates.

Third, to improve the readability, some results are presented in a table format. Additional discussions have been added to discuss the contexts, and values of this study.

Fourth, literature review, along with data analysis, is the major methodology.

Fifth, the paper has been rewritten to become more cohesive and better integrated.

Sixth, the English language has been thoroughly reviewed and edited.

Thank you again and look forward to receiving new comments.

Best, Xueming Chen and Suwei Feng

Reviewer 2 Report

Comments and Suggestions for Authors

In abstract “Even though China entered the aging society almost 20 years ago and passed elderly and disability-related laws and regulations, its transportation-related facilities and services for the elderly and disabled remain insufficient, which has seriously impacted its Transportation-Disadvantaged Populations' travel and quality of life, as well as its sustainable development. ”But “According to the National Bureau of Statistics of China, the proportion of China's elderly persons aged 65 and above jumped from 6.2% in 1995 to 10.06% in 2014.” In the section of introduction. The data in these two sentences contradict each other.

This paper does not systematically review previous studies on the U.S. and Chinese Transportation Policies and Practices for the Transportation-Disadvantaged Populations. My suggestion is that the authors should identify research gaps from previous studies by literature review, thus pointing out the contributions of the paper. 

The paper may summarize the existing policy practices and effectiveness descriptions of China and the United States in the form of a table.

The data and description in Table 1 should indicate the source of the reference.

The recommendations for the conclusion should be more specific, actionable, and implementable.

Author Response

Dear Reviewer: Thank you so much for your excellent comments.

First, the abstract has been thoroughly revised. The text has been reformatted to become more cohesive and integrated.

Second, this revision has added additional literature. In our opinion, the main research gap is the lack of comparative study between the U.S. and China, which represents the largest developed country and developing country, respectively.

Third, some comparative results were summarized in table format to improve the readability.

Fourth, Table 1 has been deleted. The related contents are included in the Discussion section.

Fixth, the recommendations for the conclusion have been revised.

Thank you again and look forward to getting your new guidance and feedback.

Best, Xueming Chen and Suwei Feng

Reviewer 3 Report

Comments and Suggestions for Authors

The application value of this manuscript should be discussed. The review recommendations for this manuscript are as follows.

1. Why does this manuscript only introduce the United States and China? What is the trend of population aging in other countries and even the whole world? Why the United States and China are representative is not mentioned in the manuscript.

2. There are only 27 literature in the manuscript, which is too few to summarize existing problem.

3. The manuscript is of little practical significance. For a review article, the focus should be on the discussion of future research trends in this field and the possible direction of policy development. However, this manuscript is only a pile of simple laws and regulations, and lacks the discussion of social, economic, medical and other challenges that may be caused by the aging trend of the population, and provides more targeted recommendations for policymakers, researchers and practitioners.

4. The focus of this manuscript is the comparison of the transportation policies and practices of China and the United States for the transportation vulnerable groups, but in the part of policy practice, there are a lot of detailed introductions of laws and regulations such as the "Twelfth Five-Year Plan" for the development of China's elderly cause. With so many lists and so little discussion, it's hard to tell whether the focus is transport-related policy.

5. It is suggested to use more charts to compare the relevant policies of the two countries, which will be more intuitive than the list of words.

Author Response

Dear Reviewer: Thank you so much for your excellent comments.

The paper has been virtually rewritten and reformatted. Here are the changes:

First, the value of this comparative study has been added in the introduction. The reasons for picking U.S. and China cases have been expounded.

Second, the number of literatures have been significantly increased.

Third, the significance of this review article has been elucidated.

Fourth, more discussions on the TDP laws and regulations have been added.

Fifth, some tables and figures have been added to improve the readability.

Thank you again and look forward to getting more guidance and feedback from you!

Best, Xueming Chen and Suwei Feng

Round 2

Reviewer 1 Report

Comments and Suggestions for Authors

Se puede publicar tal cual después de una revisión menor del idioma inglés, se sugiere que sea revisado por un hablante nativo. Si el editor así lo considera, puede publicarse después de las observaciones de los demás revisores.

Gracias

It can be published as is after a minor revision of the English language, it is suggested that it be proofread by a native speaker. If the editor considers it so, it can be published after the observations of the other reviewers.

Thank you

Comments on the Quality of English Language

Se puede publicar tal cual después de una revisión menor del idioma inglés, se sugiere que sea revisado por un hablante nativo. Si el editor así lo considera, puede publicarse después de las observaciones de los demás revisores.

Gracias

It can be published as is after a minor revision of the English language, it is suggested that it be proofread by a native speaker. If the editor considers it so, it can be published after the observations of the other reviewers.

Thank you

Author Response

Dear Reviewer #1: Thank you so much for your great suggestion and painstaking refereeing efforts. The revised article has been edited by a native English speaker.

With the best wishes, Xueming Chen and Suwei Feng.

Reviewer 2 Report

Comments and Suggestions for Authors

This article has been revised as required and it is recommended to accept.

Author Response

Dear Reviewer #2: Thank you so much for your painstaking refereeing efforts. With the best wishes, Xueming Chen and Suwei Feng.

Reviewer 3 Report

Comments and Suggestions for Authors

The revised manuscript is good and this manuscript can be accepted.

Author Response

Dear Reviewer #3: Thank you so much for your painstaking refereeing efforts. With the best wishes, Xueming Chen and Suwei Feng.